# The Reinforced Spun Concrete Poles under Physical Salt Attack and Temperature: A Case Study of the Effectiveness of Chemical Admixtures

**DOI:** 10.3390/ma13225111

**Published:** 2020-11-12

**Authors:** Romualdas Kliukas, Arūnas Jaras, Ona Lukoševičienė

**Affiliations:** Department of Applied Mechanics, Faculty of Civil Engineering, Vilnius Gediminas Technical University, Saulėtekio av. 11, LT-10223 Vilnius, Lithuania; arunas.jaras@vgtu.lt (A.J.); ona.lukoseviciene@vgtu.lt (O.L.)

**Keywords:** spun concrete, aggressive environment, physical salt attack (PSA), chemical admixtures, durability, mechanical properties, cyclic wetting and drying (CWD)

## Abstract

The present paper focused on the investigation of the effectiveness of using various chemical admixtures and their effect on the strength and deformability of the reinforced spun concrete members—the supporting poles of the overhead power transmission lines—under the unfavorable long-term combined action of the aggressive salt-saturated groundwater and the temperature changes. According to the long-term experimental program, 96 prismatic spun concrete specimens were subjected to multi-cycle (25-50-75 cycles) processing under the combined aggressive environmental conditions. It has been found that chemical admixtures which decrease the initial water-cement ratio produce a considerable positive effect on the mechanical properties of spun concrete used in hot and arid climates and exposed to physical salt attack (PSA). Superplasticizers decrease the initial water-cement ratio the most, and, due to a unique concrete compaction method used, they produce the most homogeneous and dense concrete structure. They can be recommended as most effective in increasing the durability of spun concrete used under the above-mentioned aggressive environmental conditions.

## 1. Introduction

The wide application of reinforced concrete structures in engineering leads to very different operating conditions, which are often unfavorable to concrete durability. In general, the durability of concrete depends on its resistance under the physical, chemical, or biological actions of natural or artificial origin. Concrete durability is affected by a number of factors: the quality of the concrete components (cement, water, and aggregates), the degree of concrete compaction, residual water-cement ratio (W/C), conditions of the concrete hardening, alternate wetting and drying conditions, bacterial corrosion, the attacks of various chemicals (chloride, sulphates, acids, or alkali-aggregate reactions), seawater and brine exposure, calcium leaching, freezing and thawing, carbonation, etc. Under unfavorable operating conditions, the structural changes of the already hardened concrete stone, which significantly decrease the durability of concrete structures, occurs.

General requirements to the durability of concrete and reinforced concrete structures are regulated by local design codes [1,2,3]. However, due to developing technologies and with the emergence of new concrete types, the analysis of concrete durability is still a subject of research.

In real service-life conditions, the concrete of structures is subjected, in most cases, not only to one (e.g., physical salt attack) but also to several aggressive attacks. In some cases, the joint action of several unfavorable factors at the same time may cause a more destructive effect than only the combined defects, occurring when these factors act separately. In other cases, a simultaneous action of several types of aggressive factors may lead to the gradual destruction of concrete structures, compared to the cases of a separate action of these factors. It means that the investigation of durability of concrete in the aggressive environment has to be complex in character.

The subject analyzed, i.e., spun concrete of the supporting poles of overhead power lines, in hot and arid climates is subjected not only to a single unfavorable action of temperature changes during the day-night time (up to 50–60 °C) and to the action of the aggressive groundwater, saturated with various kinds of salts, but also to the infiltration of water, saturated with salts, in concrete of the structures due to wind pressure (Figure 1). Therefore, the investigations have to be directed to the study of the resistance of concrete to the action of some particular types of the aggressive environments and to the joint simultaneous action of several unfavorable factors.

As rightly observed by O.E. Gjorv [4,5] and A. Kudzys, R. Kliukas [6], to better describe all the factors influencing the durability of concrete, a probability-based durability analysis and performance-based concrete quality control should be implemented. Unfortunately, many authors in their experiments investigate the influence of only one unfavorable factor mainly for vibrated concrete [7,8,9,10].

Focusing on the physical salts attack (PSA) on the reinforced concrete structures, dual penetration of salts into concrete or their contact with it in the areas of various climates may be observed. The first type of contact is external and appears in the colder climate zones, where salt is used as a means against icing up, in particular, for sprinkling the road surface with salt in winter. Another contact type can be observed in the warmer climate zones where the effect of salts is produced by the infiltration of salts dissolved in the groundwater and affects the concrete elements which are in contact with the soil. The present paper analyzed the supporting poles of the overhead power lines, used in the zones of dry and arid climate. Such climatic conditions occur, for example, in Asia, North Africa, and in some states of the USA. In these climatic zones, the considered negative salts’ effect is much more considerable because periodical temperature changes cause cyclic dehydration-hydration, which also accelerates the process of concrete destruction.

Salts cause a number of negative effects in contact with concrete. They decrease the concrete pH, while the acidic reaction weakens the concrete structure (increasing the size of the pores), thereby decreasing its strength. As a hygroscopic material, salt attracts and retains water in the concrete pores and capillaries. This, in turn, produces the additional damaging crystallization pressure in the cases of salt crystallization or concrete freezing. Salt also accelerates concrete carbonization, thereby contributing to the reinforcement corrosion [11], not to mention the fact that salt is one of the main sources of chlorides’ introduction into concrete, causing the reinforcement corrosion.

Due to salt crystallization taking place in the concrete pores and capillaries, the crystals formed in the pores have the ability ‘to grow’ and, in time, cause damaging crystallization pressure in concrete pores. Therefore, their size and amount, strongly depending on water-cement ratio and the technology of concrete production and the time of the salt action (the number of wetting-drying cycles) are the main influencing factors used in assessing the damaging effect of salt on concrete. For this reason, some researchers recommend the maximum W/C ratio of 0.40–0.45 [12,13,14].

It should be noted that the presence of salt in the solution improves the structure of concrete in the initial period of action, while the negative effects related to damaging crystallization pressure are noticeable only after the prolonged action of cyclic wetting and drying (CWD). The crystallization pressure is caused by the repulsive forces acting between the growing salt crystal in the CWD process and the concrete pore wall.

Currently, there are no common standard methods for testing the concrete resistance to PSA [15]. Due to its complexity, the mechanism of the salt attack is not fully disclosed and is still under theoretical and experimental investigation. A selective review of references related to salt weathering is presented by E. Doehne [16], while a book by A. Goudie and H. Viles [17] summarizing salt weathering hazards raised the awareness of the problem and broadened the field of investigation.

Apart from PSA, the second factor affecting the long-term strength of concrete is the operating temperature. The temperature accelerates not only the process of concrete paste hardening but also a whole number of physical-chemical processes taking place in the hardened concrete stone. For example, it accelerates the process of hardening and the reaction of alkaline compounds. This allows us to assume that alkali-aggregate reactions and the connected development of cracks and destruction are more intense and harmful in hot and arid regions. Only in more arid regions concrete might crack slightly less if it does not come in the direct contact with water. However, numerous cycling of dehydration and rehydration of the salts caused by temperature cycling accelerates the deterioration of concrete [3,18,19,20,21,22,23].

A general and comprehensive literature review based on the effect of the elevated temperature on concrete materials and structures is presented in Reference [18]. The detailed analysis of temperature effects on concrete in the extremely hot and arid climate was made in Reference [19,20,21,22,23].

Although there are many papers about concrete durability and the related research about various aggressive actions [4,5,6,7,8,9,10,11,12,13,14,15,16,17,18,19,20,21,22,23], the majority of them are limited to the research on vibrated concrete. The research on the durability of spun concrete subjected to PSA is seemingly lacking. There is also a lack of the field tests and indoor tests as such tests are rather expensive and very time-consuming.

Speaking about spun concrete products (including the supporting poles of overhead power lines), it should be mentioned that their production technology is specific, which results in the different concrete structure and texture and a smaller value of the residual water-cement ratio (W/C)_residual_ than those in regular vibrated concrete [24]. The performed tests have shown the high resistance and reliability of spun concrete structures [25].

One of the effective measures to improve the durability of concrete is using water-soluble chemical admixtures [26,27,28]. As shown by the research, chemical admixtures can decrease the initial water-cement ratio, not decreasing such concrete properties as workability and slump [24]. Positive qualitative mechanical characteristics of spun concrete with these admixtures are well known, but the behavior of these structures under long-term exposure to the salt environment and temperature changes have not been sufficiently investigated. There are only scarce data on the effect of the severe environment conditions on the physical and mechanical properties of spun concrete with chemical admixtures.

A series of previous papers by the authors analyzed the influence of chemical admixtures on the reinforced spun concrete elements-supporting poles of the overhead power transmission lines [24]. The present paper is focused on the investigations of the effects of various chemical admixtures on the strength and deformability parameters of the spun concrete under the combined long-term unfavorable physical actions of the aggressive salt-saturated groundwater and the great amplitude of the temperature changes during the day and night time (up to 50–60 °C). The experimental results have been carefully analyzed, and the appropriate conclusions and recommendations have been presented. This research should be continued and include other unfavorable effects on spun concrete.

## 2. The Materials and Technology Used in Producing the Experimental Specimens

### 2.1. The Materials for Making the Specimens

According to the program of the present studies, 84 spun concrete specimens of annular cross section with the external diameter of 560 mm and the wall thickness of 50 to 95 mm were made at the factory conditions. The spun specimens were made of concrete mixes based on the binders and aggregates currently used at the considered factory. Portland slag cement with the compressive strength of 40 MPa was used as a binder. Its basic physical and mechanical parameters of cement are given in Table 1.

Sand and crushed granite stone were used as fine and coarse aggregates for making the experimental specimens. The main data on the granulometric composition and physical-mechanical properties of the concrete mix aggregates are given in Table 2, Table 3 and Table 4.

To select an effective chemical admixture for increasing the physical and mechanical properties of spun concrete and creating the effective prestressed reinforced spun concrete supports of overhead power transmission lines resistant to corrosion, three-fourths of the specimens were made by using various chemical admixtures, such as superplasticizers C-3 and ‘Dofen’ (C-4), as well as acetone-formaldehyde resin ACF-3M [28]. It should be noted that the selected chemical admixtures are widely used at concrete factories in EU countries and other, as well. The superplasticizer C-3 is a synthetic product based on sulphonated naphthalene formaldehyde resin, while the superplasticizer ‘Dofen’ is an oligomeric compound based on the sodium salt and napthalenesulfonic acid. The acetone-formaldehyde resin ACF-3M is a product of the condensation of acetone and formaldehyde.

According to the case of using chemical admixtures in spun concrete described by the authors in the previous studies, the effective quantity of chemical admixtures C-3 and ‘Dofen’ is 1% of the cement mass, while the effective amount of resin ACF-3M is 0.15% of the cement mass in the concrete mixes [24]. All admixtures used were added to concrete mixes in the form of water solutions.

### 2.2. The Composition of Concrete Mixes

Low-slump concrete mixes were used for making the tubular spun concrete specimens. The composition of the concrete mixes made without using any chemical admixtures was similar to the composition of the concrete mixes used at the reinforced concrete factories for making the spun concrete supports of the overhead power transmission lines and ensuring the strength of spun concrete of 60 MPa. The compositions of the concrete mixes, including chemical admixtures, were chosen so that the slump of concrete cone of these mixes could correspond to the similar parameter of the workability of the concrete mix made without any admixtures. This was achieved by varying the water-cement ratio (W/C). The detailed composition of spun concrete mixes, including the activity of cement, granulometric composition of aggregates, and water-cement ratio, is presented in Table 5.

### 2.3. The Technology of Making Specimens

The tubular 550 mm high specimens were made by using a roller-type centrifuge for one-layer centrifuging. The metallic molds with the inner diameter of 560 mm and the length of 23,100 mm aimed at manufacturing spun concrete poles of the overhead power transmission lines were used. Special metallic diaphragms were used for forming the specimens of the specified height of 550 mm. The technological parameters of producing the experimental spun concrete elements are presented in Table 6.

A steel mold with the spun concrete was subjected to thermal treatment according to the following mode: 2 h (curing at the temperature of about +20 °C) plus 2 h (raising the temperature to +75 °C), plus 3 h (curing at the temperature of + 75 ± 5 °C), and plus 2 h (lowering the temperature). It corresponds to the mode of thermal treatment of the reinforced spun concrete poles of overhead power transmission lines. Thermal treatment of spun concrete was performed in the induction chamber, where the reinforced concrete elements were heated by heating the metal mold and the reinforcement in the electromagnetic field.

To check the strength of concrete from the initial concrete mix, simultaneously with the tubular spun specimens, the vibrated concrete cubes with the rib size of 10 cm and spun prisms were made. The latter elements were made with special attachments attached to the steel molds.

### 2.4. Specimens

In a single metal mold forty-two 550 mm high and of 50 to 95 mm thick spun concrete specimens of annular cross section were produced. Four various types of concrete mixes, including chemical admixtures and admixture free (see Table 5) were centrifuged in one the same mold (Figure 2).

A part of specimens of annular cross section were cut into prisms-segments (Figure 3), with one of their sides equal to ~200 mm, while the other side corresponded to the thickness of the annular specimen, i.e., was of 50 to 95 mm. One element of the annular cross section was cut into 8 prisms (Figure 3), two of which were left to stay under the conditions of normal temperature and humidity, while three other prisms had been wetting in the water, for example, for 25 cycles and dried in the air at the temperature of 100 °C. Three other prisms had been wetting in the salt solution for the same 25 cycles and dried in the air at the temperature of 100 °C. In this way, the reliability and comparability of the results of the experiment were ensured.

In general, 96 prisms were cut out from the spun elements of annular cross section (12 elements of annular cross section were cut) for the present experiment.

## 3. The Research Methods

In real ambient conditions concrete in reinforced concrete structures is usually exposed to more than one aggressive attack (i.e., great changes in temperature, salty groundwater, low temperature, etc.).

The total double effect of these attacks may occur:In some cases, the simultaneous action of various aggressive factors can result in slower destruction of a concrete structure than that observed in the case of their individual action.In other cases, the simultaneous action of several harmful factors on concrete results in a more destructive effect than that of the summed defects observed in concrete under the individual action of these factors.

Thus, the study of concrete durability in harmful conditions should always be complex. The research should be made in several aspects: in the direction of concrete resistance to the action of the particular types of aggressive environment and to its resistance to the complex simultaneous action of several harmful factors.

It is well-known that spun concrete, particularly the concrete in reinforced spun concrete poles of the supports of the overhead power lines, working in the countries with hot and dry climate, is subjected to great changes in the day and night air temperature, as well as to the aggressive action of the groundwater. Therefore, the present study is focused on testing the resistance of spun concrete to these two aggressive actions, also including the complex effect.

As mentioned above, the complex effect, in particular, may be ambiguous. Therefore, for complex testing of the durability of spun concrete used in hot and arid climate and subjected to the aggressive salty groundwater, the authors developed a special technique for quickly determining the corrosion resistance of spun concrete, which corresponds best to the natural aggressive environment, where the reinforced spun concrete structures, in particular, the supporting poles of overhead power transmission lines are exposed. The technique involves the alternating processes of wetting spun specimens in the water or a salt solution and drying them in the air.

According to this technique, the following modes of the aggressive environmental actions were used in testing spun concrete specimens:The alternating processes of drying the specimens in the ambient air for 8 h at the temperature of 100 °C and their wetting in water for 8 h at the temperature of 20–25 °C.The alternating processes of drying the specimens in the ambient air for 8 h at the temperature of 100 °C and their wetting in the salt solution for 8 h at the temperature of 20–25 °C.

In order to eliminate completely the additional hydration of cement, occurring in wetting the concrete stone, the complex (combined) tests were performed with spun concrete of more than one year of age.

The composition of the salt solution for creating the aggressive water environment in the laboratory conditions was chosen taking into consideration the results of the chemical analysis of the groundwater in the region of the service of the supporting poles of the overhead power transmission lines (Table 7).

It should be noted that, in real conditions, aggressive salt saturated groundwater is in contact only with one (outer) side of the concrete member. Therefore, in order to appropriately reflect a real one-sided action of the salt solution in the laboratory conditions, both ends of the prismatic specimens, as well as their lateral surfaces (along the cutting surface), were waterproofed before testing with 5 coats of polyvinyl acetate (PVA) glue. Then, the specimens were submerged in the salt solution or water (depending on the character of the aggressive environmental actions) and held there for about 24 h. The wetting process of the specimens took place in two specially equipped containers with water or the salt solution, respectively.

The specimens were dried in a specially equipped concrete drying chamber. The investigation of the resistance of spun concrete to various types of the aggressive environmental actions was performed in three stages after 25, 50, or 75 cycles of the wetting and drying, respectively.

Two prisms were kept as control specimens under the conditions of the normal ambient temperature and humidity for evaluating the level of the impact of the two considered aggressive effects on the mechanical properties of spun concrete. The distribution of prismatic specimens, depending on the testing objectives, is presented in Table 8.

When the specified number of the cycles of the alternate wetting and drying of spun concrete was completed, the main and control prismatic specimens were subjected to the short-term axial compression in the dry conditions.

To achieve more uniform inclusion of the cross section of specimens into the resistance under loading, their facing parts were inserted into special metallic casings by using the cement mortar. The specimens were tested by using a hydraulic press with the power of 5000 kN. The specimens were centered until the loads reached 25–30% of the maximal loads. For this purpose, the clock type mechanical indicators were used for measure linear strains. The loads were applied in steps amounted to 5–7% of the expected maximum strains. The tests lasted for 30–40 min, while the axial and transverse concrete strains of the prismatic specimens were measured by the strain gauges with a measuring base of 50 mm. The prism specimens were subjected to the action of the short-term axial compressive load. The specimens prepared for testing are presented in Figure 4.

## 4. The Assessment of the Testing Results

### 4.1. The Experimental Results and Assessing the Effect of the Aggressive Environment on the Mechanical Properties of Spun Concrete

The major characteristics of the cross section and the results of testing under short-term axial compression of the prismatic specimens (incl. different chemical admixtures, or free), subjected or not subjected (the control specimens) to the aggressive action, by different number of CWD, are presented in Appendix A.

The statistical analysis of the experimental data has shown that the variation coefficient of the compressive strength of the control spun concrete prisms soaked for various numbers of cycles in the water or a salt solution and dried in the air was not higher than 9% for each series and the initial modulus of elasticity was not higher than 13%. Though the number of the test specimens in each series was only three (due to experiment complexity), the above statistical indicator shows that the experiment was performed sufficiently accurately and the obtained results were reliable.

The resistance of spun concrete to the action of the aggressive environment was assessed by the dimensionless coefficients of resistance *α* and *β*. These coefficients show the degree of retaining of the initial mechanical properties of concrete under the influence of the particular aggressive environment. The resistance coefficient is determined by the ratio between the mechanical properties of concrete subjected and not subjected to the action of the aggressive environment. The coefficients *α* and *β* were determined based on two major characteristics of the mechanical properties of concrete in compression, i.e., prismatic compressive strength *f_c_* and the modulus of elasticity *E*, respectively.

Taking into consideration that various chemical admixtures were used in making the spun concrete specimens, for which testing for durability various aggressive environmental actions were employed, the three-digit indices were introduced for considering the testing specimens. Each of them indicates the type of the chemical admixture and the temperature mode, as well as the mode of wetting and drying the main specimens used.

The first index digit in the three-digit indices of the mechanical properties of spun concrete indicates the type of the chemical admixture used as follows:1.Acetone-formaldehyde resin ACF-ЗM;2.Superplasticizer ‘Dofen’;3.Superplasticizer C-3;0.The admixture-free concrete mix.

The second index digit shows the ambient temperature conditions as follows:1.Drying the specimens in the air at the temperature of 100–105 °C;0.Storing the specimens in the air at the temperature of 18–22 °C.

The third index digit shows the environmental conditions in the process of wetting the main specimens and curing the control specimens:1.Water of normal temperature (20–25 °C);2.Salt solution of normal temperature (20–25 °C);0.The air of normal humidity (the control specimens) and temperature (20–25 °C).

The resistance of spun concrete under the temperature changes was evaluated by the dimensionless coefficients as follows:(1)$$\alpha_{t}=f_{c,i,1,1}/f_{c,i,0,0},$$
(2)$$\beta_{t}=E_{c,i,1,1}/E_{c,i,0,0},$$
where *f_c,i,_*_1,1_ and *E_c_*_,*i*,1,1_ denote the prismatic compressive strength and the initial modulus of elasticity of spun concrete subjected to the cyclic actions of drying in the air under the temperature of 100–105 °C and wetting in the water of 20–25 °C; *f_c,i_*_,0,0_ and *E_c,i_*_,0,0_ indicate the prismatic compressive strength and the initial modulus of elasticity of spun concrete cured in the air of normal ambient temperature and humidity.

As mentioned above, great attention should be paid to studying the resistance of concrete used in the extreme environmental conditions to the combined effect of the aggressive action of various conditions, for example, the salt solution and temperature variations (as in the present study). The considered resistance of concrete was described by the coefficients:(3)$$\alpha_{t,c}=f_{c,i,1,2}/f_{c,i,0,0},$$
(4)$$\beta_{t,c}=E_{c,i,1,2}/E_{c,i,0,0},$$
where *f_c,i_*_,1,2_ and *E_c,i_*_,1,2_ indicate the prismatic compressive strength and the initial modulus of elasticity of spun concrete subjected to the cyclic actions of drying in the air of 100–105 °C and wetting in the salt solution at the temperature of 20–25 °C.

Using chemical admixtures in the concrete mix increases the cost of production; therefore, the efficiency of their use in concrete mixes should be determined preliminarily.

The effectiveness criterion (coefficient of effectiveness) *γ* assessing the ratio between the values of the average prismatic compressive strength of the hardened concrete made with and without the admixtures is suggested:(5)$$a)\;\;\;\gamma_{N,f_{c}}=\frac{{\bar{f}}_{c,i,0,0}}{{\bar{f}}_{c,0,0,0}};\;\;\;\;\;\;\;\;b)\;\;\;\gamma_{N,f_{c}}=\frac{{\bar{f}}_{c,i,1,1}}{{\bar{f}}_{c,0,1,1}};\;\;\;\;\;\;\;\;c)\;\;\gamma_{N,f_{c}}\;\frac{{\bar{f}}_{c,i,1,2}}{{\bar{f}}_{c,0,1,2}}.$$

In calculating this coefficient, the prismatic compressive strength of concrete stored for the same time under the same conditions (hardened in the air or soaked in water or a salt solution for the same number of cycles) were compared.

### 4.2. The Strength of Spun Concrete under Temperature Changes

The main results of evaluating the spun concrete resistance to temperature changes are presented in Table 9 and Table 10 and in Figure 5 and Figure 6. As it is demonstrated by the experimental values of the coefficients of resistance *α_t_* and *β_t_*, referring to 25, 50, and 75 cycles of alternate drying in the air and wetting in the water, a considerable negative effect of temperature changes on the mechanical properties and durability of spun concrete can be observed (Table 9 and Table 10 and Figure 5 and Figure 6).

As shown by the Figure 5 and Figure 6, after 25 cycles of alternate wetting and drying (CWD) of ordinary spun concrete, the coefficients of resistance to temperature changes make *α_t_ =* 0.79 and *β_t_* = 0.70, respectively. Thus, after 25 CWD cycles, the indicators of the mechanical properties of ordinary spun concrete are decreased by 20–30%. After 50 and 75 CWD cycles the coefficients of resistance *α_t_* and *β_t_* of ordinary spun concrete are reduced even more and make 0.72 and 0.63 and 0.68 and 0.56, respectively. The character of the changes in the data of the indicators of concrete durability, depending on the number of CWD cycles, shows that the process of microdefects’ formation in the concrete structure, caused by the temperature changes in the environment, is attenuating. This is shown by the character of the histograms presented in the Figure 5 and Figure 6.

The analysis of the experimental data has shown that chemical admixtures produce a highly positive effect on the resistance of spun concrete to temperature changes. As shown by the Figure 5, after 50 CWD cycles, the mean values of the coefficient of resistance *α_t_* of spun concrete made with admixtures vary from 0.89 to 0.91, while, for ordinary spun concrete, this coefficient makes 0.72.

After 75 CWD cycles, the coefficient of resistance *α_t_* for concrete with admixtures ranges from 0.77 to 0.85, while, for spun concrete without any admixtures, it is by 10–20% smaller and makes 0.68. As shown by the Figure 5 and Figure 6, the temperature gradient and CWD decrease the initial modulus of elasticity of spun concrete more strongly than the strength of spun concrete.

As shown by the Figure 6, after 75 CWD cycles, the coefficient of resistance *β_t_* for spun concrete with admixtures is between 0.68 and 0.75, while, for ordinary concrete, it equals 0.56. It can be observed that with the increase in the number of CWD cycles, the effectiveness of using chemical admixtures is increased considerably. It is confirmed by the fact stated above [24] that chemical admixtures produce a positive effect on structural formation of spun concrete in the process of its curing.

The negative effect on the mechanical concrete properties is produced by temperature changes and the alternate swelling/shrinkage in the process of the cyclic drying and wetting. Their actions cause the formation of micro cracks in concrete, in which its intensity actually depends on the gradient of the temperature in the CWD process.

According to the data presented by A. J. Al-Tayyibm et al. [19], the heating/cooling of concrete affects its durability performance as it loses up to 27 and 32% of its compressive and flexural strength, respectively. For all the tested specimens (under 80 °C for 24 h), the greatest loss of strength could be observed after 30 cycles. However, in the research performed by V. Korovyakov et al. [20], the optimal content of concrete is offered. The study by S. A. Al-Saleh [21] has shown that, when the temperature changes to ~16 °C, the compressive strength decreases, while the slump increases.

It should be mentioned that a positive effect of concrete wetting on the strength and deformability of spun concrete is caused by the additional hydration of the cement stone due to the additional humidity obtained by it, which is required for this process.

As shown by the histograms presented in Figure 5 and Figure 6, the values *α_t_* and *β_t_* characterizing the resistance of spun concrete to temperature changes only slightly depend on the type of the chemical admixture used. This makes the process of identifying the most effective admixture more difficult. It can only be mentioned that, based on both indicators of the considered resistance of spun concrete, the most effective admixtures in the considered case appeared to be C-З and ‘Dofen’.

### 4.3. The Resistance of Spun Concrete to a Complex Effect of Temperature Changes and the Salt Attack

In assessing the resistance of spun concrete to a complex effect of temperature changes and the salt solution, the data matching the above described data on the influence of chemical admixtures on the mechanical characteristics of spun concrete have been obtained [24].

The data obtained in testing spun concrete, subjected to 25, 50, and 75 cycles of alternate drying in the air and wetting in the salt solution, have shown that admixtures actually produce the same positive effect on the considered indicator of spun concrete resistance. It can only be mentioned that, in the considered case, the admixtures C-3 and ‘Dofen’ appeared to be most effective. This is shown by the experimental values of the coefficients of resistance *α_t,c_* according to (3) and *β_t,c_* according to (4), given in Table 11 and Table 12 and in Figure 7 and Figure 8, and characterizing the resistance of spun concrete to the complex effect of temperature changes and the salt attack.

Comparing the data given in the histograms in Figure 5, Figure 6, Figure 7 and Figure 8, it can be observed that in the case of a complex cyclic effect of temperature changes and a salt attack, the decrease in the mechanical indicators’ values of ordinary spun concrete is not faster than that found under the condition of a particular aggressive effect of temperature changes on the considered concrete. This can be explained as follows.

In the case of a complex effect of a salt solution and temperature changes, the negative effect of the latter, shown by the formation of micro cracks in concrete, is smoothed by a positive effect of new formations of salt crystals in micro cracks and pores on the strength and deformability of concrete.

Thus, in the case of the above complex aggressive effect of the environment, the corrosion of concrete characterized by the crystallization of salts in the concrete pores with the consequent pressure of the edges of the growing crystals on the walls of the pores, in the considered case (after 50 CWD cycles) produced a positive rather than a negative effect on the mechanical properties of ordinary spun concrete. This means that the corrosion of the third type ‘cured’ the damaged structure of ordinary spun concrete caused by cyclic temperature changes.

The most detailed analysis of stresses caused by salt crystallization was performed by G. W. Scherer [29,30]. There, four cases of crystallization pressure—capillary rise and evaporation, as well as cyclic wetting and drying, precipitation of ettringite, and hydration of cement—were examined. The broader analysis of the considered problem may be found in References [31,32,33,34,35,36,37,38,39]. In these studies, the crystallization process and damaging crystallization pressure were analyzed from the theoretical and experimental perspective.

However, in the case of a larger amount of the CWD cycles (*n* > 50), when the process of micro cracks’ formation in concrete caused by the cyclic temperature changes actually stops, a positive effect of the corrosion of the third type on the strength and deformability of concrete (by curing the defects) turns into a negative effect because the pressure on the walls of the pores and micro cracks exerted by the edges of the growing crystals can be observed. This was confirmed by the results of the tests performed by using prisms when, after 75 cycles of their wetting in the salt solution, the coefficient of resistance *α_t,c_* decreased considerably.

In the case of a complex effect of the salt solution and the temperature changes on spun concrete made with chemical admixtures, the effect of the corrosion of the third type on its mechanical properties obtained in the described case differed from that obtained for ordinary concrete. As shown in Figure 5 and Figure 7, after 50 CWD cycles, the coefficient *α_t,c_* of spun concrete made by using chemical admixtures decrease more intensely than the coefficient *α_t_.* This shows that the crystallization of salts in the case of concrete swelling-shrinkage causes the development of the stress of stretching in the concrete pores.

This phenomenon can be explained as follows. The formation of the micro cracks in spun concrete made by using chemical admixtures caused by the cyclic temperature changes in the environment is not as intense as that observed in ordinary spun concrete (Figure 5). At the same time, the process of the defects’ formation in the structure of spun concrete made with chemical admixtures dies in the shorter period of time than a similar process in ordinary spun concrete. Due to this, in the denser concrete, like spun concrete with new micro cracks made by using chemical admixtures, the negative effect of the pressure of the edges of the growing salt crystals on the pores’ walls and micro cracks can be observed earlier and is more evident. As mentioned above, the number of micro pores and micro capillaries in spun concrete depends on the composition of the concrete mix, of which an object is made, and, particularly, on the initial water-cement ratio (Table 5). In the considered case, it was the concrete mix without admixtures (W/C)_initial_ = 0.37, while, with ACF-3M, it is equal to 0.37 and with ‘Dofen’ and C-3 it is equal to 0.30 and 0.29, respectively.

Based on the above mentioned facts, it can be concluded that corrosion resistance of spun concrete under the complex aggressive effect of the salt solution and temperature changes can hardly be determined only by the coefficients *α_t,c_* according to (3) and *β_t,c_* based on (4) since both these influencing factors (temperature changes and concrete corrosion of the third type) are interrelated.

### 4.4. The Effectiveness of Chemical Admixtures in Concrete Mixes

The use of chemical admixtures in making the objects of spun and vibrated concrete has been discussed in one of the papers [24], where, based on the experimental results, it had been stated that using chemical admixtures in concrete mixes for making spun concrete structures allows for decreasing water consumption of a concrete mix, not decreasing its workability. This helps to obtain concrete with a denser structure and, therefore, with a higher strength. Moreover, admixtures accelerate the structural formation of concrete in the conditions of its curing and thereby improve its physical and mechanical properties in the process of production.

As a result, a considerable economy of cement and a shorter time of curing the concrete can be obtained, which, in turn, allows for the economy of power resources, faster reuse of metal molds and smaller labor expenditure at the plants of reinforced concrete structures’ production [24]. In the previous studies [24], it had been stated that a positive effect of chemical admixtures on spun concrete was particularly pronounced at an early age of concrete.

This paper presents the research results, allowing us to summarize the effectiveness of using chemical admixtures in spun concrete at the age of up to 1.5 years, as well as in concrete soaked in the water or the salt solution for 25, 50, or 75 cycles and dried in the air at 100 °C, and their effect on its durability.

As shown by the data given in Table 13, it can be seen that the values of the coefficient of effectiveness γ of using chemical admixtures in the concrete mixes when spun concrete after the curing of half a year (*n* = 25), a year (*n* = 50) or 1.5 years (*n* = 75) hardened under normal ambient temperature and humidity were decreasing and seemed to depend on the initial water-cement ratio of the concrete mix (see Table 5). Therefore, after 1.5 years, it is equal to ~1.0 for concrete with the ACF-3M admixture and ~1.16 for concrete with the admixtures C-3 and ‘Dofen’.

Considering the data given in Appendix A, it can be observed that both for concrete without admixtures and with admixture ACF-3M (in which initial water-cement ratio (W/C)_initial_ ≈ of 0.35 to 0.38) the strength increases by ~16% per year, while, for concrete with admixtures C-3 and ‘Dofen’, the strength increases by ~2–5% per 1.5 years.

These relationships are similar to those obtained for spun concrete mixes with chemical admixtures and for spun concrete soaked in water or a salt solution (Table 13) and dried at 100 °C.

The compression stress of these types of concrete after their subjection to the action of the aggressive environment for 75 cycles and reaching the age of 1.5 years exceeds the stress of concrete without any admixtures by 40–50% when admixtures C-3 and ‘Dofen’ are used and only by ~10% when the admixture ACF-3M is used.

When turned into the economic effect, the effective service time of spun concrete supporting poles of the overhead electric power transmission lines together with the repair and replacement of the poles, the sums of milliards of euros could be obtained. The producing cost of a single supporting pole of 110 kV and its mounting in the overhead electric power transmission line in Lithuania is over 4.5 thousand Euros.

## 5. Conclusions

It has been determined that, after 25, 50, and 75 cycles of concrete wetting in the water and drying in the air, the temperature gradient (80 °C) produce a considerable negative effect on the mechanical properties of spun concrete. Depending on the number of the wetting-drying cycles, the compression strength of concrete without admixtures is decreased by 20–30%, while its modulus of elasticity is decreased by 30–40%. However, the compression strength of spun concrete with superplasticizers C-3 and Dofen, which reduce the initial water-cement ratio, is decreased by 10–15%, while its modulus of elasticity is decreased by 20–25%. The compression strength and the modulus of elasticity of spun concrete with the ACF-3M admixture, which only slightly reduces the initial water-cement ratio, are decreased to a greater extent–by 15–20% and 25–30%, respectively.It has been found that under the complex action of temperature changes and a salt solution, complex and important processes are taking place in the structure of spun concrete. First, the fading process of the concrete structure’s destruction caused by the cyclic temperature changes takes place. Second, the salt solution contributes to the improvement of the concrete structure in the initial period of the investigated environmental effect. Under the action of a long-term complex effect of temperature changes and the salt solution, the positive effect of concrete corrosion of the third type turns into a negative effect. After 75 cycles of the complex effect of temperature and the salt solution, the compression strength of spun concrete without admixtures decreased by 35%, while the modulus of elasticity–by 40%. For concrete with admixtures, these mechanical properties decreased to a lesser extent, by 25–35%.The coefficients of effectiveness (5) describing the effectiveness and economic use of chemical admixtures, decreasing the initial water-cement ratio in the concrete mix, show that the considered admixtures improve the mechanical properties of spun concrete used in dry and hot climatic conditions by up to 1.5 times.

## Figures and Tables

**Figure 1 materials-13-05111-f001:**
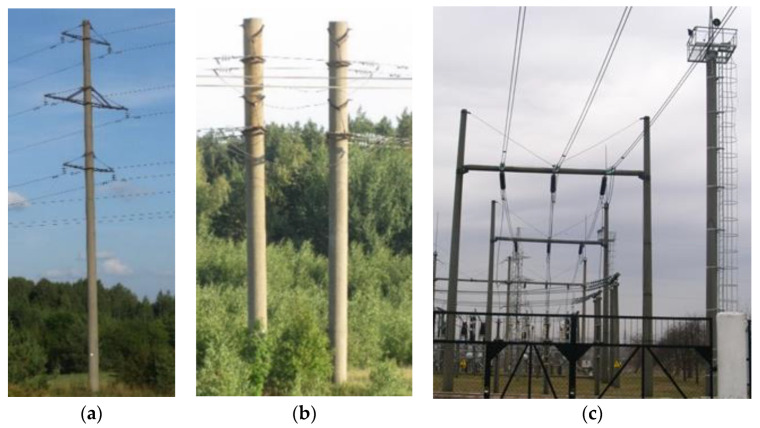
The pillars of the overhead electric power lines: (**a**) monopole; (**b**) bipole; (**c**) electric power distribution substations.

**Figure 2 materials-13-05111-f002:**
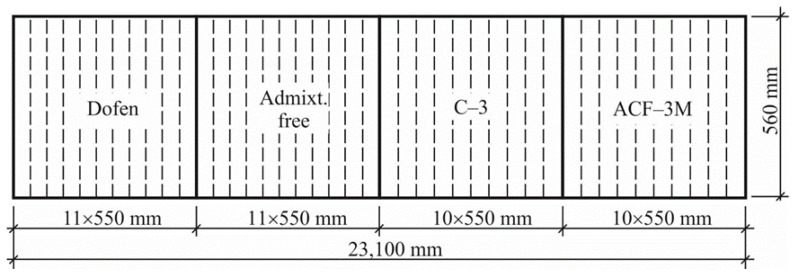
The arrangement of the elements of annular cross section made of concrete mixes of various compositions along the mold.

**Figure 3 materials-13-05111-f003:**
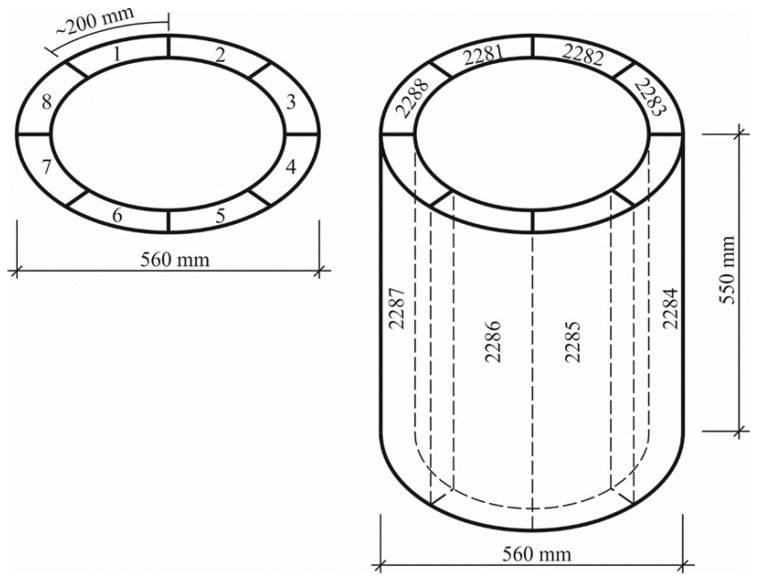
A schematic view of cutting annular cross section elements into prisms (in vertical position, including codes of prism specimens).

**Figure 4 materials-13-05111-f004:**
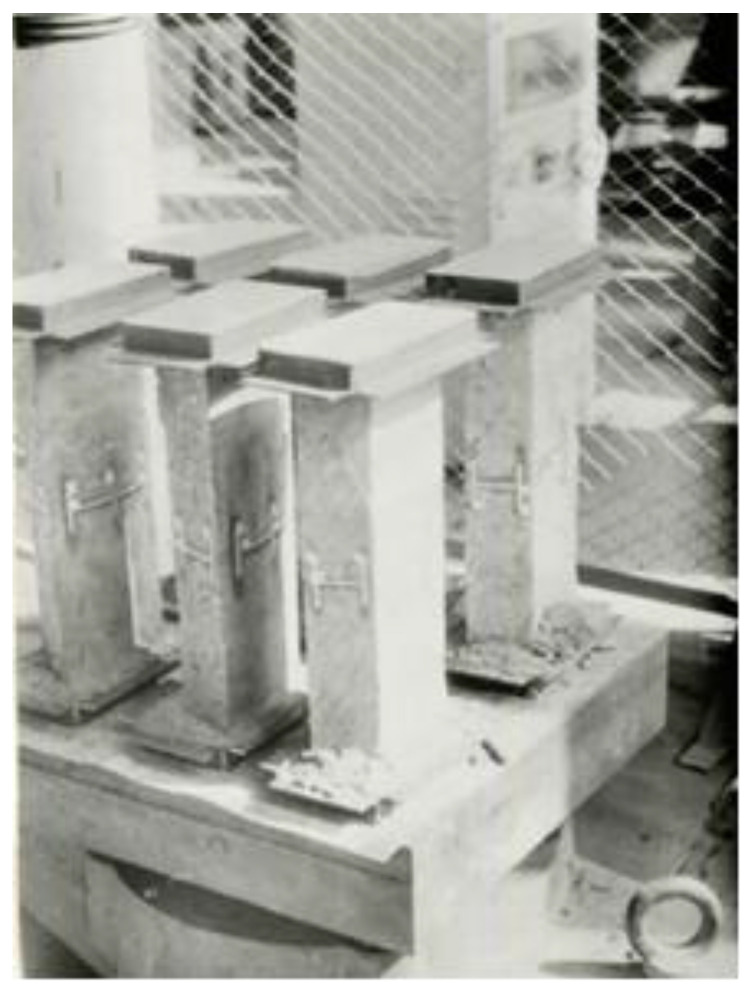
Prism specimens prepared for testing.

**Figure 5 materials-13-05111-f005:**
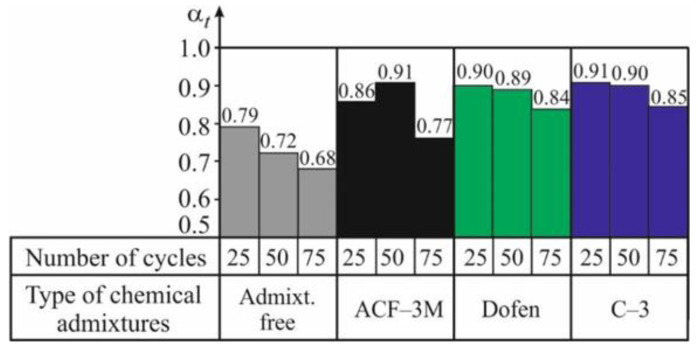
The graph of the relationship between the coefficient of resistance *α_t_* of spun concrete based on (1) and the number of cycles of its alternate drying in the air at 100–105 °C and wetting in the water at 20–25 °C.

**Figure 6 materials-13-05111-f006:**
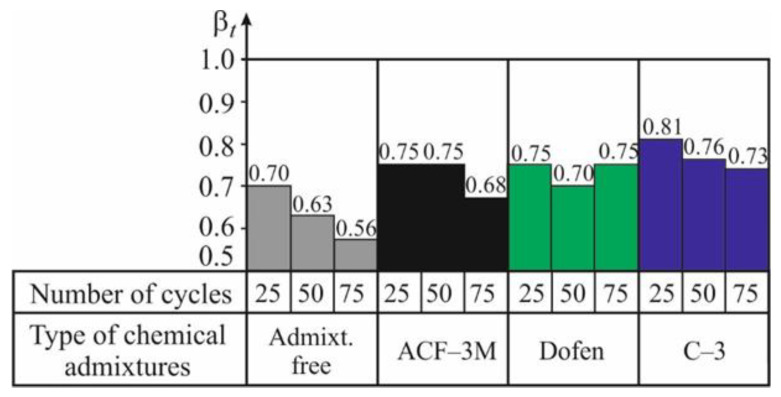
The graph of the relationship between the coefficient of resistance *β_t_* of spun concrete based on (2) and the number of cycles of its alternate drying in the air at 100–105 °C and wetting in the water at 20–25 °C.

**Figure 7 materials-13-05111-f007:**
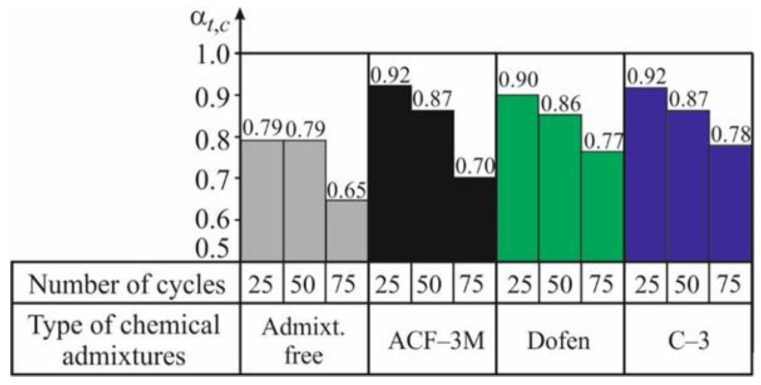
The graph of the relationship of the coefficient of resistance *α_t,c_* of spun concrete according to (3) on the number of cycles n of its alternate drying in the air at 100–105 °C and wetting in the salt solution at 20–25 °C.

**Figure 8 materials-13-05111-f008:**
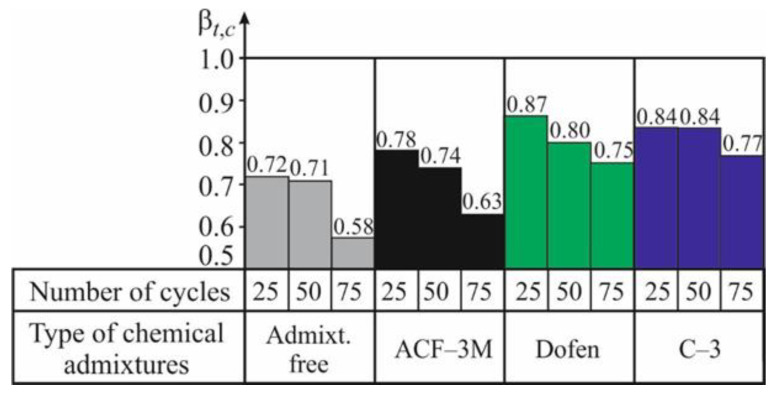
The graph of the relationship of the coefficient of resistance *β_t,c_* of spun concrete according to (4) on the number of cycles n of its alternate drying in the air at 100–105 °C and wetting in the salt solution at 20–25 °C.

**Table 1 materials-13-05111-t001:** The main physical-mechanical parameters of Portland slag cement.

Parameter	Unit	Value
Normal consistence of cement grout	%	26.2
Initial setting time	h	3.25
Final setting time	h	4.75
Fineness of grinding	%	85
Volumetric density	kN/m^3^	26.5
Strength of samples of hard consistency at the age of 3 days:		
- under compression	MPa	21.5
- under bending	MPa	3.8

**Table 2 materials-13-05111-t002:** The granulometric composition of sand.

Dimensions of Sieve Holes d (mm)	Partial Residues (g)		Average Value of Partial Residues		Whole Residues (%)
	I-1000 g	II-1000 g	(g)	(%)	
2.5	˗	˗	˗	˗	˗
1.25	15	12	13.5	1.35	1.35
0.63	48	53	50.5	5.05	6.4
0.314	152	149	150.5	15.05	21.5
0.14	735	742	738.5	73.85	95.3
0.14	50	44	47	4.7	100

**Table 3 materials-13-05111-t003:** The granulometric composition of crushed granite stone.

Dimensions of Sieve Holes d (mm)	Partial Residues (g)		Average Value of Partial Residues		Whole Residues (%)
	I-3000 g	II-3000 g	(g)	(%)	
25	311	309	310	10.3	10.3
20	1099	1101	1100	36.7	47.0
15	1061	1059	1060	35.3	82.2
10	480	480	480	16	98.3
5	29	31	30	1	99.3
2.5	20	20	20	0.7	100

**Table 4 materials-13-05111-t004:** The main physical-mechanical parameters of sand and crushed granite stone.

Material	Parameter	Unit	Amount
1. Sand	Fineness modulus	-	1.24
	Volumetric density	kN/m^3^	13.5
	Volumetric compacted density	kN/m^3^	16.4
	Humidity ratio	%	2.2
	Amount of contaminants	%	0.2
2. Crushed granite stone	Volumetric density	kN/m^3^	15.1
	Volumetric compacted weight	kN/m^3^	17.0
	Amount of contaminants	%	0.1
	Grade of crushed stone (by fineness)	kPa	100

**Table 5 materials-13-05111-t005:** Composition of spun concrete mixes.

No of Mix	Admixture Type	Amount of Admixtures (% of Cement Mass)	Amount of Components of Mix (kg/m^3^)				Water–Cement Ratio
			Cement	Sand	Crushed Stone	Water	
1	Dofen	1.0	565	400	1280	170	0.30
2	Admixt. free	-	565	400	1280	208	0.37
3	C-3	1.0	565	400	1280	164	0.29
4	ACF-3M	0.15	565	400	1280	209	0.37

**Table 6 materials-13-05111-t006:** Technological parameters of concrete spinning.

No of Spinning Stage	Spinning Rate (rpm)	Spinning Time (min)
1	50–80	3
2	150	1
3	200	1
4	300	1
5	420–442	15

**Table 7 materials-13-05111-t007:** Chemical composition of 1 m^3^ solution of salts.

Salts	Salt Content (kg)
Magnesium sulphate (MgSO_4_ × 7H_2_O)	2.89
Magnesium chloride (MgCl_2_ × 6H_2_O)	34.98
Calcium chloride (CaCl × 2H_2_O)	49.26
Sodium chloride (Na Cl)	239.75
Potassium bicarbonate (KHCO_3_)	0.24
Ammonium chloride (NH_4_Cl)	0.24
Total salt content:	327.3
Water content:	672.7

**Table 8 materials-13-05111-t008:** Distribution of prismatic spun concrete specimens by testing objectives.

Types and Purposes of Sub-Studies	Number of Cycles of CWD	Number of Specimens				
		Main Specimens (Incl. Admixtures, and Free)				Control Specimens (Admixt. Free)
		C-3	Dofen	ACF-ЗM	Admixt. Free	
1) Study of the resistance of spun concrete to cyclic variation in temperature	25	3	3	3	3	2 × 3 × 4 = 24
	50	3	3	3	3	
	75	3	3	3	3	
2) Study of the resistance of spun concrete to the complex action of salt solutions and cyclic variation in temperature	25	3	3	3	3	
	50	3	3	3	3	
	75	3	3	3	3	
Total amount:		18	18	18	18	24

**Table 9 materials-13-05111-t009:** The experimental values of the coefficient of resistance *α_t_* (based on (1)), depending on the number of cycles *n* of concrete alternate drying in the air and wetting in the water.

Type of Chemical Admixtures	The Average Compressive Strength of Concrete ${\bar{f}}_{c,i}$ (MPa)			The Average Compressive Strength of the Control (Not Soaked) Specimens ${\bar{f}}_{c,i}$ (MPa)			The Coefficients of Resistance α_t_ (Based on (1))		
	If the Number n of Cycles of Wetting in Water and Drying in the Air at 100 °C						If the Number n of Cycles of Wetting in Water and Drying in the Air at 100 °C		
	25	50	75	25	50	75	25	50	75
Admixt. free	39.0	40.0	38.8	49.1	55.6	57.1	0.79	0.72	0.68
ACF-3M	43.0	54.4	44.2	50.1	59.6	57.4	0.86	0.91	0.77
Dofen	56.8	68.4	55.6	62.8	77.1	66.2	0.90	0.89	0.84
C-3	60.9	63.1	57.5	66.8	70.0	67.6	0.91	0.90	0.85

**Table 10 materials-13-05111-t010:** The experimental values of the coefficient of resistance *β_t_* (based on (2)), depending on the number of cycles *n* of concrete alternate drying in the air and wetting in the water.

Type of Chemical Admixtures	The Average Initial Modulus of Elasticity ${\bar{E}}_{c,i}\;(\mathrm{GPa})$			The Average Initial Modulus of Elasticity of the Control (Not Soaked) Specimens, ${\bar{E}}_{c,i}\;(\mathrm{GPa})$			The Coefficients of Resistance *β_t_* (Based on (2))		
	If the Number *n* of Cycles of Wetting in Water and Drying in the Air at 100 °C						If the Number *n* of Cycles of Wetting in Water and Drying in the Air at 100 °C		
	25	50	75	25	50	75	25	50	75
Admixt. free	20.3	20.0	19.5	29.0	31.8	33.6	0.70	0.63	0.56
ACF-3M	24.6	26.6	19.8	32.6	35.5	29.1	0.75	0.75	0.68
Dofen	27.2	28.8	27.8	36.2	41.0	37.0	0.75	0.70	0.75
C-3	29.6	28.8	26.1	36.5	38.0	35.7	0.81	0.76	0.73

**Table 11 materials-13-05111-t011:** The experimental data of coefficient of resistance *α_t,c_* (based on (3)), depending on the number of the cycles *n* of wetting the concrete in the salt solution.

Type of Chemical Admixtures	The Average Compressive Strength of Concrete ${\bar{f}}_{c,i}$ (MPa)			The Average Compressive Strength of the Control (Not Soaked) Specimens ${\bar{f}}_{c,i}$ (MPa)			The Coefficients of Resistance *α_t,c_* (Based on (3))		
	If the Number n of Cycles of Wetting in Salt and Drying in the Air at 100 °C						If the Number n of Cycles of Wetting in Salt and Drying in the Air at 100 °C		
	25	50	75	25	50	75	25	50	75
Admixt. free	38.8	44.1	37.2	49.1	55.6	57.1	0.79	0.79	0.65
ACF-3M	46.0	51.8	40.2	50.1	59.6	57.4	0.92	0.87	0.70
Dofen	56.4	66.4	51.0	62.8	77.1	66.2	0.90	0.86	0.77
C-3	61.3	60.8	52.7	66.8	70.0	67.6	0.92	0.87	0.78

**Table 12 materials-13-05111-t012:** The experimental data of coefficient of resistance *β_t,c_* (according to (4)), depending on the number of the cycles *n* of wetting the concrete in the salt solution.

Type of Chemical Admixtures	The Average Initial Modulus of Elasticity ${\bar{E}}_{c,i}\;(\mathrm{GPa})$			The Average Initial Modulus of Elasticity of the Control (Not Soaked) Specimens, ${\bar{E}}_{c,i}\;(\mathrm{GPa})$			The Coefficients of Resistance *β_t,c_* (Based on (4))		
	If the Number *n* of Cycles of Wetting in Salt and Drying in the Air at 100 °C						If the Number *n* of Cycles of Wetting in Salt and Drying in the Air at 100 °C		
	25	50	75	25	50	75	25	50	75
Admixt. free	20.9	22.6	15.0	29.0	31.8	33.6	0.72	0.71	0.58
ACF-3M	25.4	26.4	18.3	32.6	35.5	29.1	0.78	0.74	0.63
Dofen	31.6	32.6	27.8	36.2	41.0	37.0	0.87	0.80	0.75
C-3	30.5	31.9	27.5	36.5	38.0	35.7	0.84	0.84	0.77

**Table 13 materials-13-05111-t013:** The coefficient of effectiveness of using chemical admixtures in spun concrete mixes based on (a), (b), (c), according to number of cyclic wetting and drying (CWD).

Type of Chemical Admixtures	Coefficient of Effectiveness *γ_N,fc_* (Based on 5 a, b, c)								
	5 a			5 b			5 c		
	Number of Cyclic Wetting and Drying (CWD)								
	25	50	75	25	50	75	25	50	75
Admixt. free	1.0	1.0	1.0	1.0	1.0	1.0	1.0	1.0	1.0
ACF-3M	1.02	1.07	1.01	1.10	1.36	1.14	1.19	1.17	1.08
Dofen	1.28	1.39	1.16	1.46	1.71	1.43	1.45	1.51	1.37
C-3	1.36	1.26	1.18	1.56	1.58	1.48	1.58	1.38	1.42

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
