# Peer review of "The Reinforced Spun Concrete Poles under Physical Salt Attack and Temperature: A Case Study of the Effectiveness of Chemical Admixtures"

_materials, 2020, doi:10.3390/ma13225111_

Round 1

Reviewer 1 Report

The paper entitled "The Reinforced Spun Concrete Poles under Physical Salt Attack and Temperature – a Case Study of the Effectiveness of Chemical Admixtures " has potential to be interesting for scientific community.

Generally, the text of the paper is well written. 

The methods of research are solid. The presentation is technically correct. The interpretation of results is adequate.

Specific comments are as follows:

Title
The title clearly describes the paper.

Abstract
It is written: ... C-3 and Dofen can be recommended as the most effective admixtures to be used for increasing the durability of spun concrete ... What is C-3, Dofen? Abstract contains the names of chemical admixtures. It says nothing to potential readers. There should be a chemical name of the admixture and possibly the name in parentheses. This is explained in the paper, but trade names should not be used in the abstract.

Introduction
Introduction is interesting, contains quite a lot of references to literature, and finally contains the specific purpose of the authors' research. However, I get the impression that the theoretical part is too long. Information such as that contained in sections 127-135 should rather be in part of the Discussion that is actually missing from this paper.

Methodology
The methods of research are solid and the materials are well documented but:
- Why plasticized blast-furnace cement was used as a binder? Please explain.
-   Line 186-190; 197-204: Please fill in the properties of the chemical admixtures selected for testing. There is no information other than the chemical name.

4.The Assessment of the Testing Results

The results included in all tables were not statistically analyzed (standard deviation, coefficient of variation). Concrete is a heterogeneous material and the spread of results should be given in the tables.

Some of the results can be presented in a more interesting form, because the extensive tables with the results themselves are not legible and the reader gets lost in the mass of information.

5. Results and discussion

This point should only be called Discussion, because the Section 4 describes Results in detail.

The current results look like a slightly technical report. Please complete the discussion after each study. There are practically no references to literature or other studies. Authors should payed more attention to comparison of their results with similar publications, which would be mostly recommended.
In addition, the authors throughout the paper write about the influence of chemical admixtures without separating them in terms of their chemical nature. Each admixture is from a completely different chemical group. This parameter was not analyzed at any point of paper. You should not write only trade names, eg. Dofen and others. Please explain why each admixture influenced in such a way, e.g. increased durability? How did it affect the cement, its hydration, the formation of the C-S-H phase, etc.? There are no such analyzes here.

Conclusions
No conclusions at work. Short, appropriate conclusions are expected, including data and results.

According my suggestion the paper needs major restructuration.
I recommend the paper to publish after major revisions.

Author Response

Dear Reviewer,                                                                                                       2020.11.04.

Thank you for your valuable comments that have contributed to improving the quality of the article. Article has been revised and a number of amendments and corrections (incl. Abstract, Body text-Results and Conclusions presented) have been made.

Answers and comments according specific comments:

Comments:

Answers and comments:

Abstract. It is written: ... C-3 and Dofen can be recommended as the most effective admixtures to be used for increasing the durability of spun concrete ... What is C-3, Dofen? Abstract contains the names of chemical admixtures. It says nothing to potential readers. There should be a chemical name of the admixture and possibly the name in parentheses. This is explained in the paper, but trade names should not be used in the abstract.

Comments accepted.  Abstract rewritten. The trademarks are rejected. While the chemical names are explained in main text (155-160). The chemical formulas are not published.

Introduction. Introduction is interesting, contains quite a lot of references to literature, and finally contains the specific purpose of the authors' research. However, I get the impression that the theoretical part is too long. Information such as that contained in sections 127-135 should rather be in part of the Discussion that is actually missing from this paper.

Comments accepted.  Introduction revised and shortened. A part of information (former 129-135) moved in main text (398-403).

Methodology
The methods of research are solid and the materials are well documented but:

1) Why plasticized blast-furnace cement was used as a binder? Please explain.

2)    Line 186-190; 197-204: Please fill in the properties of the chemical admixtures selected for testing. There is no information other than the chemical name.

Comments accepted.

1) Portland slag cement was used as a binder, because this cement is mainly used in concrete factories for producing RC members investigated.

2) The plasticizing chemical admixtures that allows reducing the W/C ratio, without reducing workability of concrete were selected. These admixtures are widely used in concrete factories.

The Assessment of the Testing Results

1) The results included in all tables were not statistically analyzed (standard deviation, coefficient of variation). Concrete is a heterogeneous material and the spread of results should be given in the tables.

2)  Some of the results can be presented in a more interesting form, because the extensive tables with the results themselves are not legible and the reader gets lost in the mass of information

Comments accepted.

1) The results of statistical analysis presented (299-304).

2) A part of results data (Tables 9-11) moved in to Supplementary Materials.

5. Results and discussion

1) This point should only be called Discussion, because the Section 4 describes Results in detail.

The current results look like a slightly technical report. Please complete the discussion after each study. There are practically no references to literature or other studies. Authors should payed more attention to comparison of their results with similar publications, which would be mostly recommended.
2) In addition, the authors throughout the paper write about the influence of chemical admixtures without separating them in terms of their chemical nature. Each admixture is from a completely different chemical group. This parameter was not analyzed at any point of paper. You should not write only trade names, eg. Dofen and others. Please explain why each admixture influenced in such a way, e.g. increased durability? How did it affect the cement, its hydration, the formation of the C-S-H phase, etc.? There are no such analyzes here.

Comments accepted.

1) Appropriate corrections in the body text were made (398-403), 450-455). It should be noted, that experimental data published are mostly related with ordinary vibrated concrete. It is very scarce experimental data, related with experimental investigations of spun concrete.

2) The utilizing plasticizing chemical admixtures that allows reducing the W/C ratio (as one of most important factors related to durability of concrete), without reducing workability of concrete were the main reason of investigation.

The broader information requested is presented in our previous paper [24]. However, the chemical nature of admixtures has not be analyzed.

Conclusions
No conclusions at work. Short, appropriate conclusions are expected, including data and results

Comment accepted.

Conclusions revised and corrected (527-549).

Best Regards,

Reviewer 2 Report

The paper treats about The Reinforced Spun Concrete Poles under Physical Salt Attack and Temperature.

The paper is well written and it is worthy of publication in the Journal.

A minor review is required.

Page 5 line 180. I guess that Authors would mean “compressive strength” instead of “activity”

Page 9: the quality of figure 4 is too poor. Please substitute it with a high quality figure.  

Page 10: the quality of figure 5 is too poor. Please substitute it with a high quality figure.  

Author Response

Dear Reviewer,                                                                                                       2020.11.04.

Thank you for your valuable comments that have contributed to improving the quality of the article. Article has been revised and a number of amendments and corrections (incl. Abstract, Body text-Results and Conclusions presented) have been made.

Comments and Suggestions for Authors

Comments and Suggestions:

Answers and comments:

Page 5 line 180. I guess that Authors would mean “compressive strength” instead of “activity”

Comment accepted.

Corrected (however, both titles are acceptable).

Page 9: the quality of figure 4 is too poor. Please substitute it with a high quality figure.  

Comment accepted.

Figure 4 rejected.

Page 10: the quality of figure 5 is too poor. Please substitute it with a high quality figure.  

Comment accepted.

Figure 5 replaced.

Best Regards,

Reviewer 3 Report

The authors presented a detailed study on the effects of cyclic temperature variation and a combination of temperature change / salt solution on the mechanical properties of spun concrete, and discussed the positive contributions of the inclusion of chemical admixtures to the concrete durability. In general, the paper is well written and fits the scope of the Materials journal well. However, the authors should consider the following modifications for improvement of the paper quality.

  1. The authors provided a detailed literature review, however, it talked about too many topics including concrete durability (affecting factors, requirements), salts (negative effects), salt crystallization, concrete resistance to PSA, joint effects of unfavorable factors, temperature effects, concrete production technology and chemical admixtures. With so many aspects discussed, the introduction part seemed too lengthy and scattered, and was difficult for readers to follow. Meanwhile, some topics of great relevance were not comprehensively discussed, such as chemical admixtures. I would suggest the authors make the introduction more concise with fewer topics, but provide more detailed reviews on several selected topics.
  2. I would suggest the authors should consider moving the tables 9-11 into the supplemental section, given that the critical data was repeated in tables 12-15 and the paper was already lengthy. By doing this, the critical information would not be compromised and the paper would become more concise.
  3. In Lines 409-417 and 431-434, the authors stated that the decrease in the coefficients of resistance α & β suggests the process of micro-defect formation (micro-cracks) in the concrete structure. I don’t see why the authors can draw such conclusions based on the current experimental results. Is this a hypothesis that the authors proposed to explain the change? Does literature suggest the same thing? If so, the authors should make it clear. I think this is a very important statement, because the authors also talked about this when discussing the positive effects of salt crystallization later.

Author Response

Dear Reviewer,                                                                                                       2020.11.04.

Thank you for your valuable comments that have contributed to improving the quality of the article. Article has been revised and a number of amendments and corrections (incl. Abstract, Body text-Results and Conclusions presented) have been made.

Comments and Suggestions for Authors

Comments and Suggestions:

Answers and comments:

1. The authors provided a detailed literature review, however, it talked about too many topics including concrete durability (affecting factors, requirements), salts (negative effects), salt crystallization, concrete resistance to PSA, joint effects of unfavorable factors, temperature effects, concrete production technology and chemical admixtures. With so many aspects discussed, the introduction part seemed too lengthy and scattered, and was difficult for readers to follow. Meanwhile, some topics of great relevance were not comprehensively discussed, such as chemical admixtures. I would suggest the authors make the introduction more concise with fewer topics, but provide more detailed reviews on several selected topics.

Comment accepted.

The Introduction is revised and shortened. Some paragraphs rejected and some (former 129-135) moved to the body text (398-403).

2. I would suggest the authors should consider moving the tables 9-11 into the supplemental section, given that the critical data was repeated in tables 12-15 and the paper was already lengthy. By doing this, the critical information would not be compromised and the paper would become more concise.

Comment accepted.

Tables 9-11 moved in to Supplementary Materials.

3.  In Lines 409-417 and 431-434, the authors stated that the decrease in the coefficients of resistance α & β suggests the process of micro-defect formation (micro-cracks) in the concrete structure. I don’t see why the authors can draw such conclusions based on the current experimental results. Is this a hypothesis that the authors proposed to explain the change? Does literature suggest the same thing? If so, the authors should make it clear. I think this is a very important statement, because the authors also talked about this when discussing the positive effects of salt crystallization later.

Comment partially accepted.

In the previous our researches [24], (studying microstructure of spun concrete), the radial micro capillaries, formed due to the removal of excess water from concrete in spinning process, were fixed. The broader information requested presented in our previous paper [24].

Best Regards,

Round 2

Reviewer 1 Report

I accept after authors' corrections.

Reviewer 3 Report

The authors made suggested modifications and the paper quality was significantly improved. I think the paper can be accepted for publication.